# Simultaneously Monitoring Whole Corneal Injury with Corneal Optical Density and Thickness in Patients Undergoing Cataract Surgery

**DOI:** 10.3390/diagnostics11091639

**Published:** 2021-09-07

**Authors:** Tzu-Han Hsieh, Hun-Ju Yu, I-Hui Yang, Ren-Wen Ho, Yu-Ting Hsiao, Po-Chiung Fang, Ming-Tse Kuo

**Affiliations:** 1Department of Medical Education, Kaohsiung Chang Gung Memorial Hospital, Kaohsiung City 83301, Taiwan; b101102016@tmu.edu.tw; 2Department of Ophthalmology, Kaohsiung Chang Gung Memorial Hospital and Chang Gung University College of Medicine, Kaohsiung City 83301, Taiwan; angelayu@cgmh.org.tw (H.-J.Y.); rubyihy@cgmh.org.tw (I.-H.Y.); wen6530@cgmh.org.tw (R.-W.H.); yuting1008@cgmh.org.tw (Y.-T.H.)

**Keywords:** corneal optical density, corneal thickness, Pentacam, cataract surgery, corneal wound injury

## Abstract

To pursue the least corneal implication during cataract surgery, this study aimed to monitor corneal wound injury after cataract surgery with a novel method. The prospective cohort study involved thirty-two patients, who were assessed by a Scheimpflug tomography AxL^®^ (Oculus GmbH, Wetzlar, Germany) via the following two kinds of indices: whole corneal optical density (COD) and corneal thickness (CT), two weeks before and one month after cataract surgery. The results of the COD revealed that corneal annuli 0.0–2.0 mm and 2.0–6.0 mm, and the average and maximal values at the incisional site significantly increased postoperatively. Also, the anterior and central stroma of 0.0–2.0 mm, and all three depths of 2.0–6.0 mm, increased remarkably after the operation. For the CT, all ranges of diameters plus incisional sites showed significant increases postoperatively. Furthermore, we analyzed the differences (delta) of COD and CT between pre- and post-operation, and found significant correlations between the delta of COD and the delta of CT, regarding annuli 0.0–2.0 mm, 2.0–6.0 mm, and 6.0–10.0 mm, but no correlation at the incisional site, with either average density or maximal density, was detected. We concluded that whole COD and CT, especially at the central zones of the cornea (annulus < 6 mm), are both valuable parameters in the assessment of corneal damage post-cataract surgery, and are independent indices at the incisional site.

## 1. Introduction

Cataract surgery is the most common ocular surgery in the world. In adults over 50 years of age, the global prevalence of cataract is about 50% [1]. During the outbreak of COVID-19 in the world, it is estimable that each month of lockdown caused a reduction of about 50,000 cataract operations in Italy, one of the most affected countries [2]. Nowadays, cataract surgery has proven to be one of the most cost-effective healthcare interventions. Cataract-related visual impairment has effects not only on daily life activities, but also on one’s psychological wellbeing [3]. With the dramatic increments and high life quality in the elderly in developed countries, ophthalmologists aspire for the least harm to the cornea during any sophisticated procedure. Among these, phacoemulsification with the clear corneal incision is the most well-performed surgery in recent decades, but has the risk of mechanical trauma and thermal damage [4]. Therefore, before the following modern technique comes out, we search for feasible, objective, and precise parameters to monitor corneal damage and recovery after cataract surgery in clinical practice.

Corneal thickness (CT) has been well known to be an endothelial damage index after surgery [5,6,7]. Previously, it was detected via ultrasound pachymetry, specular microscopy, or in vivo confocal microscopy, providing crucial information on corneal injury after cataract surgery [5,6,8]. However, the above modalities examine only the corneal injury locally instead of wholly. Both ultrasound pachymetry and in vivo confocal microscopy contact a patient’s cornea during the examination, making invasiveness and infection concerns unavoidable. Recently, non-contact imaging techniques, using light scattering-based systems to assess CT, have emerged [8,9]. However, they have been criticized for their inaccuracy compared with ultrasound pachymetry [8,10].

Some studies report that surgical damage or accidental contact with the endothelium results in corneal edema and swelling through the mechanism of local inflammation, epithelial permeability increment, loss of cytoskeletal structures, endothelial cell loss, and endothelial pump dysfunction [7,11,12,13]. With corneal edema, the corneal thickness and opacification increase [6,14]. Corneal opacification, caused by a thermal burn or iatrogenic trauma during cataract surgery, is related to edema or collagen condensation, which may compromise the visual outcome [15]. Thus, we plan to simultaneously monitor corneal optical density (COD) and corneal thickness (CT) before and after cataract surgery, to detect corneal injury and the recovery condition by Pentacam.

Pantacam AxL^®^ (Oculus GmbH, Wetzlar, Germany), equipped with a rotating Scheimpflug camera, claims to provide excellent measurement towards the anterior eye segment [16]. It can assess not only the CT, but also the COD of the whole corneal dimension.

Similarly to many other cataract surgeons worldwide, we routinely follow up with patients who undergo cataract surgery one day, one week, and one month after the operation in our hospital. Follow-up at four weeks after cataract surgery is a critical moment to decide the end of follow-up or continuous care for these patients. Thus, we aim to elucidate the corneal injury prudently at the 4-week follow-up, for patients who undergo cataract surgery, via the simultaneous identification of COD and CT changes with a whole corneal dimension and, additionally, at the incisional site, to observe the surgical adverse effects directly.

To our knowledge, this is the first report to monitor and compare the two parameters in corneal surgery. We believe that this study’s findings will help compare the performance of varied cataract instruments, cataract technologies, and the self-monitoring of cataract surgery manipulation for a surgeon.

## 2. Materials and Methods

### 2.1. Study Design

This is a prospective cohort study with seventy-five patients who participated initially. The phacoemulsification cataract surgeries were performed by three experienced ophthalmologists in October 2018 at the Department of Ophthalmology in Kaohsiung Chang Gung Memorial Hospital, a tertiary healthcare hospital in Taiwan. Each participant acquired informed consent, and all measurements were performed in accordance with the Declaration of Helsinki and approved by the local ethics committee (IRB certification no.: 201800899B0). Inclusion criteria were patients over 40 years old with a cataract-induced visual disability, such as refractive errors unsuccessfully corrected by spectacles, and consented to receive cataract surgery. Subjects were excluded if pregnant, breastfeeding, had poorly controlled diabetic mellites (HbA1C > 12%) or retinopathy.

### 2.2. Surgical Procedure

Patients with nuclear opacification degrees equal to 3 or more, based on the classification of lens opacities classification system III (LOCS III), were included in the study. All the patients manually received small-incision phacoemulsification cataract surgery via an Infiniti phacoemulsification (Alcon Laboratories, Fort Worth, TX, USA) with standardized settings under topical anesthesia. Before being covered with surgical draping, 10% and 5% povidone-iodine solutions were used to sterilize periocular and ocular surfaces, respectively. After several eyedrops of 0.5% proparacaine hydrochloride and 0.5% levofloxacin, the surgical procedure was performed according to the following routine steps: 2.75 mm × 0.5 mm clear corneal incision, aqueous fluid replacement with ophthalmic viscosurgical devices, continuous curvilinear capsulorhexis with capsular forceps, hydrodissection and hydrodelineation with balanced salt solution, nuclear disassembly with the divided and conquer technique, cortical removal, implantation of intraocular lens, removal of ophthalmic viscosurgical devices, and wound sealing with hydration. After the completion, intracameral cefuroxime 1 mg/0.1 cc was injected. Before pressure bandage with gauze and patching with a metal shield, 0.5% levofloxacin eyedrops, and 0.3% tobramycin and 0.1% dexamethasone ophthalmic ointment were instilled.

### 2.3. Pre- and Post-Operative Assessment

Participants underwent a complete preoperative examination two weeks before surgery and the same postoperative examination one month after surgery, including intraocular pressure, refractometry, keratometry, visual acuity, CT, and COD. The data of COD and CT were obtained by the Pentacam AxL^®^ (Pentacam AxL, Oculus GmbH, Wetzlar, Germany) examination.

Pentacam AxL^®^ provided a three-dimensional scan of the anterior eye segment, which measured light backscattering in grayscale units (GSUs) on a scale of 0 to 100, to quantify the whole COD at concentric annuli (0.0 to 2.0 mm, 2.0 to 6.0 mm, 6.0 to 10.0 mm, and 10.0 to 12.0 mm) and three depths (anterior 120 μm, central stroma, and posterior 60 μm). In the whole CT analysis, the program values averaged on annuli rings from radius 0.0 to 5.0 mm (annuli 0 to 10 mm with 2 mm interval).

### 2.4. Sample Size Determination

The sample size was determined by a free online calculator (https://sample-size.net/sample-size-study-paired-t-test/ (accessed on 12 October 2018)) supported by the Clinical and Translational Sciences Institute of the University of California, San Francisco. According to the prior result of corneal thickness change at the wound from the first ten qualified patients, we estimated the sample size by adopting the significance level (α) as 0.05, the desired power (1-β) as 0.8, the effect size of 18, and the standard deviation of the corneal wound’s thickness change of 33.2. Accordingly, the estimated sample size was at least 29 subjects for this study.

### 2.5. Statistical Analysis

Statistical analysis was performed with Excel 2019. The Kolmogorov–Smirnov test was used to test for the normal distribution of the data. For data following a normal distribution, the 1-sided Student t-test was used for paired values, and the Pearson correlation coefficient was performed by an online calculator (https://www.answerminer.com/calculators/correlation-test/ (accessed on 10 September 2020)) [17]. If the normal distribution cannot be fitted, the 1-sided Wilcoxon signed-rank test and Spearman correlation coefficient would be used. A *p*-value of less than 0.05 was considered statistically significant.

For correlation analysis between the concentric CODs and CTs of the whole cornea, a parallel comparison was used by densitometry annuli 0.0 to 2.0 mm to pachymetry 0.0 mm, densitometry 2.0 to 6.0 mm to pachymetry 4.0 mm, and densitometry 6.0 to 10.0 mm to pachymetry 8.0 mm. For surgery-related factors analysis, we used the data from the area of the incisional corneal wound on the densitometry and pachymetry map (Figure 1). The delta value was defined as a postoperative value minus a preoperative value of the same parameter. Furthermore, the increment rate was described as a postoperative value minus a preoperative value and then divided by a preoperative value for the same parameter.

## 3. Results

### 3.1. Participants

Among the seventy-five patients who initially participated, forty subjects were excluded for analysis because they did not pass the automatic quality assessment of the examination by the Pentacam AxL^®^ (Oculus GmbH, Wetzlar, Germany), pre- or post-operatively. We summarized the quality assessment results in Appendix A. Another three patients were excluded because their CODs at the surgical site could not be detected. Therefore, thirty-two patients were enrolled for analysis in this study in the end, including 15 males and 17 females, with a mean age of 67 years old. The baseline demographic data are shown in Table 1. Overall, 62.5% of the patients received phacoemulsification cataract surgeries in their right eye. The average axial length was 24.67 mm, while the anterior chamber depth was 3.04 mm. The mean pre- and post-operatively uncorrected distance visual acuities were log MAR 0.89 and log MAR 0.31, respectively (*p* < 0.000).

There was no significant difference in concentric COD between genders, in either the range of diameter or incisional site (annuli 0.0–2.0 mm, *p* = 0.384; annuli 2.0–6.0 mm, *p* = 0.440; annuli 6.0–10.0 mm, *p* = 0.419; annuli 10.0–12.0 mm *p* = 0.263, respectively, max COD at incisional site, *p* = 0.458) (Figure 2a). Also, there was no significant difference in concentric CT between genders (central CT, *p* = 0.253; annuli 2.0 mm, *p* = 0.229; annuli 4.0 mm, *p* = 0.282; annuli 6.0 mm, *p* = 0.357, annuli 8.0 mm, *p* = 0.365, respectively, incisional site, *p* = 0.297) (Figure 2b).

The preoperative COD of the total diameter at the total depth significantly increased with age (*p* = 0.0003) (Figure 3a). Also, it increased with the diameter, as a concave up curve. Through the curve, the minimal COD was at annuli 2.0 to 6.0 mm. Besides, the COD in the anterior layer had the predominant role within the total depth, and the posterior layer had the least character (Figure 3b). Preoperatively, there was a significant correlation between age and average COD at the incisional site (*r* = 0.437, *p* = 0.012) (Figure 3c), but no correlation with the maximal COD (*r* = 0.226, *p* = 0.215) (Figure 3d).

### 3.2. Changes of the CODs and CTs at Concentric Annuli and Incisional Site

To evaluate the surgical effect, we compared the preoperative and postoperative values of the whole COD and CT of each subject. In the COD, the annuli 0.0 to 2.0 mm (*p* = 0.002), 2.0 to 6.0 mm (*p* = 0.001), average (*p* = 0.002) and maximal values (*p* = 0.004) at incisional site significantly increased after cataract surgery (Figure 4a). To further investigate the change in each depth, the concentric CODs were significantly increased in the anterior (*p* = 0.004) and central layer (*p* = 0.005) of 0.0 to 2.0 mm, and in all three depths (anterior, *p* = 0.001; central, *p* = 0.021; posterior, *p* = 0.025) of 2.0 to 6.0 mm (Figure 4b). On the other hand, the concentric CTs increased significantly after cataract surgery, including 0.0 mm, 2.0 mm, 4.0 mm, 6.0 mm, 8.0 mm, and the incisional site (*p* value all < 0.000) (Figure 4c).

### 3.3. Correlation between CODs’ and CTs’ Changes

In the correlation analysis between the changes in concentric CODs and those of corresponding CTs, the delta values (postoperative values minus preoperative values) were all significantly correlated, including annuli 0.0 to 2.0 mm (*r* = 0.5067, *p* = 0.0031), annuli 2.0 to 6.0 mm (*r* = 0.4390, *p* = 0.0119), and annuli 6.0 to 10.0 mm (*r* = 0.4967, *p* = 0.0038). However, neither the delta of the average density (*ρ* = 0.1778, *p* = 0.3303) nor the delta of the maximal density (*ρ* = 0.2395, *p* = 0.1867) was correlated with the delta of the thickness at the incisional site (Figure 5). Therefore, we further analyzed the increment rate between concentric CODs and CTs. However, neither the increment rate of average optical density (*r* = 0.069, *p* = 0.706) nor that of maximal optical density (*r* = 0.157, *p* = 0.390) had significant correlation with the increment rate of CTs (Table 2).

## 4. Discussion

In our study, the COD is the lowest in the central zone and the highest in the periphery divided by the radial zones. When separated by depth, the anterior layer displays the highest COD, compared to the central and posterior layers. Ní Dhubhghaill et al., in 2014, [18] assessed normative values for CODs, and their results are in alignment with our findings, making COD a reliable measurement in our study. However, the mean COD over the total 12 mm diameter area was 19.74 ± 3.89 GSU in their study [18], which is much lower than our results (24.24 ± 5.57 GSU). Another difference is that previous studies showed that COD correlated positively with age only in the peripheral zone (annuli > 6 mm) [7,19]. We found that age had significant positive correlations with the COD in all four radial zones. We think age distribution plays a critical role. In our study, the mean age was 67.0 ± 9.2 years (range, 44.4–82.7 years), corresponding to cataract being age-related. Whether COD in the central zones (diameter < 6 mm) follows the same pattern needs future studies to confirm, especially in the aged group.

Both the postoperative average and maximal CODs at the incisional site were significantly higher than the preoperative ones, which means that the surgical intervention does interfere with the corneal wound thoroughly and continually, even one month after cataract surgery. On the other hand, as divided by the radial zone, significant postoperative COD increases were in the central zone (central circle 0–2 mm, and concentric annuli 2–6 mm), rather than the periphery zone (concentric annuli 6–10 mm, and 10–12 mm). We speculate that the central zone is the most transparent in the entire cornea, and a tiny increase in the COD can reach a statistical difference. The corneal incisional wound may cause a tremendous injury, with only causal increases in the CODs in the peripheral zone. However, the CODs in the large annular areas against the incisional sites in the periphery zone might compensate for the induced changes in the incisional wound, resulting in no statistical difference.

Previously, there were several methods to obtain corneal thickness (CT) data. Ultrasound pachymetry is the gold standard, by directly touching the cornea to get the most precise values. Specular microscopy can assess the corneal endothelial loss, and in vivo confocal microscopy can further evaluate the change in cells and nerves in different layers of the cornea [5,6,8]. However, corneas are vulnerable after surgery and cannot stand any risk of re-damage or infection. Both pachymetry and in vivo confocal microscopy come into contact with a patient’s cornea during the examination, not to mention the above modalities only examine the confined area of the cornea.

Non-contact pachymetry, such as anterior segment optical coherence tomography (ASOCT) and Pentacam tomography, uses light scattering-based imaging systems [8,20]. It possesses the advantages of being non-invasive, user-independent, time-effective, and with a comprehensive assessment. Our study uses Pentacam to measure CT, and confirms that older people have thinner corneas [21,22]. Nonetheless, CT measured by light scattering has been criticized for inaccuracy [9,10,23]. It is reported that ASOCT and Pentacam tomography may underestimate the central CT in normal eyes [8,10]. Pentacam tomography tends to overestimate in acutely edematous corneas, compared with ultrasound pachymetry. We assume that the false-increase CTs in the acutely damaged cornea may contribute to the significance of all the findings over the preoperative and postoperative period in our study. It may raise a doubt as to whether non-contact corneal pachymetry can provide a precise examination. From another perspective, the pre-op underestimation and postoperative overestimation in the CT measurement by Pentacam tomography might amplify the hidden corneal injury and facilitate better postoperative monitoring.

The relationship between COD and CT has been investigated in healthy subjects, where no correlation was found [19,24]. However, no previous study has explored their relationship in corneal injuries or disorders, which is one of our primary research focuses in this study. We compared the two variables by subtracting preoperative and postoperative values to obtain the delta values for each subject. The results showed that the changes (delta values) in the CODs, and those of the CTs were all significantly correlated in annuli 0 to 2 mm, 2 to 6 mm, or 6 to 10 mm. The finding corroborates that COD and CT are both relevant and vital parameters for monitoring the corneal status entirely after cataract operations.

Against our speculation, there was no correlation between COD and CT at the incisional site. Moreover, neither the average nor the maximal COD increment rate had a significant correlation with the increment rate of the corresponding CT, indicating that the CODs’ and CTs’ parameters were independent to the corneal injury indices at the incisional site. We suppose that this may be due to the thermal burn during phacoemulsification, which greatly increases COD, but only slightly increases CT at the primary incisional wound. In addition, some techniques adopted in cataract surgery, such as divided and conquer, to deliver a high phaco power or hydroseal to close the incision wounds, may explain this discrepancy [8]. Whether cataract surgery affects stroma hydration persistently [25,26] or facilitates corneal haze formation transiently needs further research to elaborate.

How long before the cornea recovers after undergoing phacoemulsification remains controversial. Fukuda et al. showed that corneal thickness had no significant change two weeks after the surgery [25]. However, Suzuki et al. inspected corneal volume and found that peripheral cornea edema persisted after one month of follow-up [27]. Our study showed that CODs had significant abnormality at the central annulus and incisional site, even one month after cataract surgery, while the CTs in all the radial zones significantly increased. It seems that CT presents with better sensitivity, but COD has better specificity towards assessing phacoemulsification-related corneal damage. Also, the recovery timeline varies individually, depending on each patient’s corneal condition, the surgeon’s experience, the extent of damage during the manipulation, the postoperative care, etc. Therefore, we promote monitoring the CTs and CODs before and after corneal interventions, with a series of examinations, for the early detection and follow-up of the surgical outcomes in the future.

The limitation of this study is that the indices are measured only with Pentacam. We do not compare it with other Scheimpflug cameras or anterior segment optical coherence tomography that is currently available on the market. Further studies may be needed to elucidate whether other Scheimpflug cameras present the same pattern of densitometry and pachymetry as Pentacam. Another pitfall is the difficulty in passing all of the automatic quality assessments of Pentacam AxL^®^, which led to the exclusion of many subjects who were initially recruited for analysis. Despite the instrument claiming to have a short measuring time, within 2 s, this examination may still be difficult for certain patients. In elderly patients, many unpredictable conditions may result in the challenge of performing this examination, such as achalasia, dry eye disease, hearing disability, or poor cooperation, which may require experienced technicians to overcome.

## 5. Conclusions

This study demonstrates that whole CODs and CTs, especially at the central zones of the cornea (annuli < 6 mm), detected by Pentacam tomography, are highly valuable indices for corneal injury monitoring after cataract surgery. Therefore, in the era of precision medicine, we believe that the simultaneous and personalized monitoring COD and CT, via Pentacam tomography, is a promising, sensitive and specific modality for determining corneal injury or recovery.

## Figures and Tables

**Figure 1 diagnostics-11-01639-f001:**
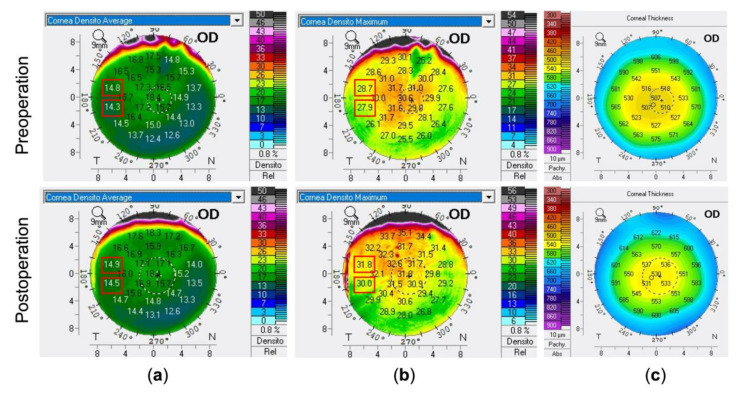
Densitometry and pachymetry maps. (**a**) Average densitometry map for obtaining the mean optical densities at different zones of cornea. (**b**) Maximal densitometry map for observing the maximal optical densities at different zones of cornea. (**c**) Pachymetry map for assessing the mean corneal thickness at varied sites of cornea. For the temporal approach of cataract surgery, the values highlighted with the two red rectangles were averaged to obtain the corneal optical density at the incisional site.

**Figure 2 diagnostics-11-01639-f002:**
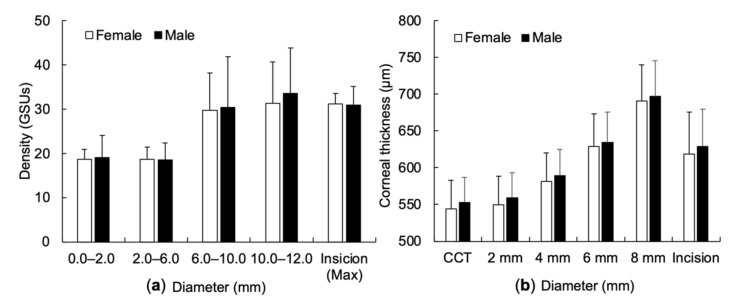
Preoperative corneal optical density and corneal thickness with sex. (**a**) Corneal optical density between sex in either radial diameter or incisional site. (**b**) Corneal thickness between sex in either radial diameter or incisional site. Density = corneal optical density; GSUs = grayscale units; CCT = central corneal thickness.

**Figure 3 diagnostics-11-01639-f003:**
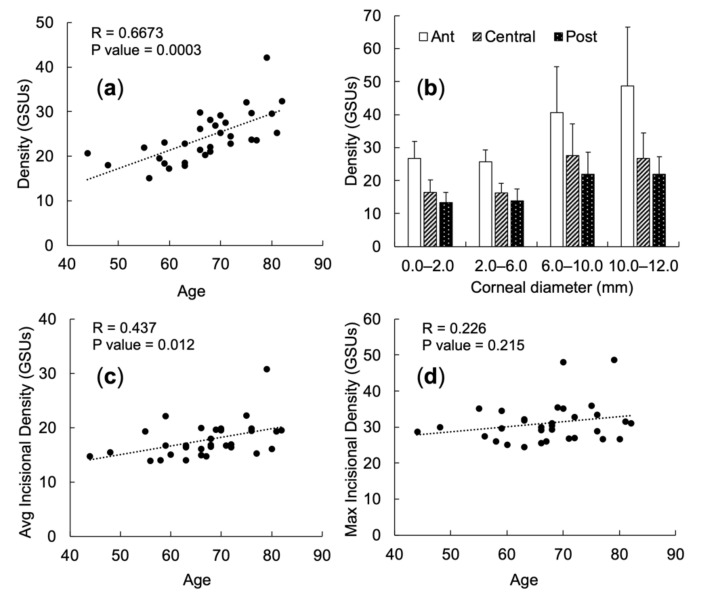
Preoperative corneal optical density of corneal diameter and incisional site. (**a**) The correlation between corneal optical density (COD) and age. (**b**) Comparing the CODs among the anterior, central stroma, and posterior layers of the cornea. (**c**) Correlation between age and average COD at the incisional site. (**d**) Correlation between age and maximal COD at the incisional site. Density = corneal optical density; GSUs = grayscale units; ant = the anterior layer of cornea (anterior 120 µm); post = the posterior layer of cornea (posterior 60 µm); central = the central stroma of cornea between ant and post.

**Figure 4 diagnostics-11-01639-f004:**
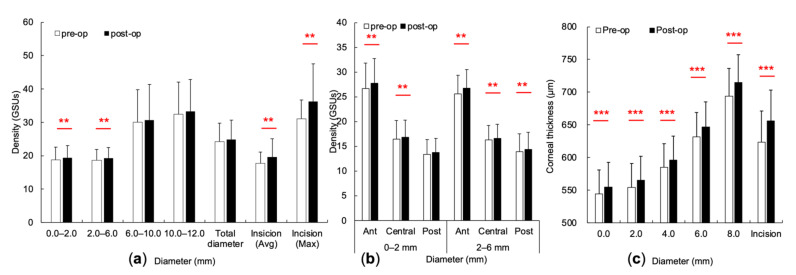
Changes in corneal optical density and corneal thickness regarding concentric diameter and incisional site after cataract surgery. (**a**) Comparison of the corneal optical density (COD) at the concentric diameter and incisional site. (**b**) The concentric CODs in different layers of the cornea at annuli 0.0 to 2.0 mm and 2.0 to 6.0 mm. (**c**) The concentric corneal thickness at zones 0.0 mm, 2.0 mm, 4.0 mm, 6.0 mm, 8.0 mm, and incisional site. ** *p* < 0.01, and *** *p* < 0.001, compared to the preoperative values.

**Figure 5 diagnostics-11-01639-f005:**
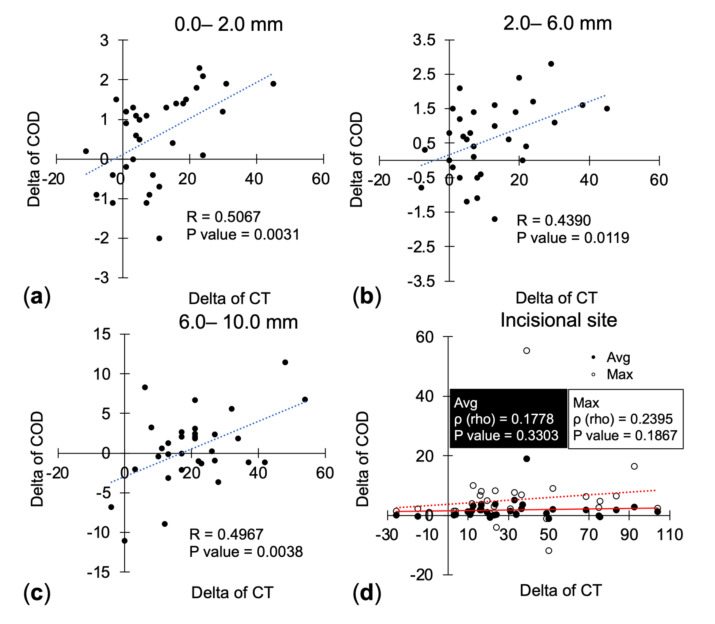
Correlation between the change in corneal optical density and the change in corneal thickness after cataract surgery. The delta value of corneal optical density and corneal thickness at (**a**) the annulus 0.0 to 2.0 mm, (**b**) the annulus 2.0 to 6.0 mm, and (**c**) the annulus 6.0 to 10.0 mm, (**d**) the incisional site. COD = corneal optical density; CT = corneal thickness.

**Table 1 diagnostics-11-01639-t001:** Demographic data.

	Overall (N = 32)
Age	67.03 ± 9.18
Sex	
Female	17/32 (53.13%)
Male	15/32 (46.87%)
Laterality	
OD	20/32 (62.5%)
OS	12/32 (37.5%)
Axial length (mm)	24.67 ± 2.28
Anterior chamber depth (mm)	3.04 ± 0.47
Pre-op bare vision (logMAR)	0.89 ± 0.88
Pre-op best-corrected visual acuity (logMAR)	0.55 ± 0.77
Post-op bare vision (logMAR)	0.31 ± 0.66

**Table 2 diagnostics-11-01639-t002:** Correlations between corneal optical density and corneal thickness at the incisional site.

	COD (GSUs)	CT (μm)	r	*p* Value
	Average COD	CT		
Pre-op	17.78 ± 3.33	623.34 ± 53.66	−0.090	0.626
Post-op	19.65 ± 5.40	656.27 ± 67.04	0.152	0.408
Increment rate	0.099 ± 0.160	0.056 ± 0.041	0.069	0.706
	Maximal COD	CT		
Pre-op	31.09 ± 5.57	623.34 ± 53.66	−0.080	0.663
Post-op	36.31 ± 11.28	656.27 ± 67.04	0.217	0.232
Increment rate	0.172 ± 0.296	0.056 ± 0.041	0.157	0.390

Note: increment rate = (post-op value−pre-op value)/pre-op value; COD = corneal optical density; CT = corneal thickness; *r* = Pearson correlation coefficient.

## Data Availability

Data available on request from authors.

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
