# Peer review of "Simultaneously Monitoring Whole Corneal Injury with Corneal Optical Density and Thickness in Patients Undergoing Cataract Surgery"

_diagnostics, 2021, doi:10.3390/diagnostics11091639_

Round 1
Reviewer 1 Report
English language needs editing (grammatically and speeling errors)
The abstract needs to follow aims, methods, results and conclusion
Initially 74 patients participated in the study but 42 were excluded. Please elaborate.
In the inclusion criteria, refractive error is mentioned. Please elaborate.
Also please explain the surgical procedure details and the density of cataract.
Please explain what kind of study is presented. It seems like an observational type.
Please add the sample size calculation
Please add axial length and anterior chamber depth in the demographic data table.
Please acknowledge the limitations of the current study such as non-randomisation, etc.
The manuscript requires major revision.
Author Response
Q1 English language needs editing (grammatical and spelling errors).
A1
The revised manuscript has been re-edited by a physician proficient in English.
Q2 The abstract needs to follow aims, methods, results, and conclusion.
A2
We have added a statement about the design of the study as a prospective cohort study and adjusted the abstract structure following aims, methods, results, and conclusion (page 1, lines 16-29).
Q3 Initially 75 patients participated in the study but 43 were excluded (page 2, line 74). Please elaborate
A3
Thanks for this reminder. We have added some statements to describe why 43 subjects were excluded for analysis (page 2, lines 85-91). In addition, we provide the supplementary files to clarify the quality assessment results of Pentacam AxL® preoperatively (Supplementary Table S1) and postoperatively (Supplementary Table S2). Only thirty-five patients passed the automatic quality assessment of this Scheimpflug system in both preoperative and postoperative examinations (page 9, lines 322-324). Another three patients were excluded because their CODs at the surgical site cannot be detected. Therefore, thirty-two patients were enrolled for analysis in this study in the end.
Table S1. Preoperative quality assessment of Pentacam AxL® examination
|
Number (eye) |
Percent |
|
|
OK |
46 |
61.3% |
|
AXL SNR Error |
6 |
8.0% |
|
Lid closure ! |
6 |
8.0% |
|
Alignment (Z) Error |
5 |
6.7% |
|
Blinking Error ! |
5 |
6.7% |
|
Unsteady Fixation |
3 |
4.0% |
|
AXL SNR Error Single Peak |
2 |
2.7% |
|
AXL Align. (XY) Error |
1 |
1.3% |
|
Data Gaps ! |
1 |
1.3% |
|
Total |
75 |
100.0% |
Table S2. Postoperative quality assessment of Pentacam AxL® examination
|
Number (eye) |
Percent |
|
|
OK |
50 |
66.7% |
|
Lid closure ! |
8 |
10.7% |
|
Alignment (Z) Error |
4 |
5.3% |
|
Unsteady Fixation |
4 |
5.3% |
|
Alignment (XY) Error |
2 |
2.7% |
|
Blinking Error ! |
2 |
2.7% |
|
3D Model Deviation ! |
2 |
2.7% |
|
AXL Align. (XY) Error |
1 |
1.3% |
|
Data Gaps ! |
1 |
1.3% |
|
Blinking Error / Nose Shadow! |
1 |
1.3% |
|
Total |
75 |
100.0% |
Q4 In the inclusion criteria, refractive error is mentioned. Please elaborate.
A4
We thank the reviewer for pointing out this detail. We found that this was an incomplete statement and we have corrected this statement as “Inclusion criteria were patients aged over 40 years old with cataract-induced visual disability such as refractive errors unsuccessfully corrected by spectacles, and consented to receive cataract surgery.” (page 2, lines 82-84).
Q5 Also please explain the surgical procedure details and the density of cataract.
A5
According to the suggestion, we have revised this manuscript in Materials and methods (pages 2-3, lines 92-107) as follows: Patients with nuclear opacification degree equal to 3 or more, based on the classification of Lens Opacities Classification System III (LOCS III), were included in the study. All the patients received manually 2.75 mm ´ 0.5 mm small-corneal incision phacoemulsification cataract surgery via an Infinity phacoemulsificator (Alcon Laboratories, Fort Worth, Texas, USA) with standardized settings under topical anesthesia. Before covered with surgical draping, 10% and 5% povidone-iodine solutions were used to sterilize periocular and ocular surface, respectively. After several eyedrops of 0.5% proparacaine hydrochloride and 0.5% levofloxacin, surgical procedure was performed according to the following routine steps: clear corneal incision, aqueous fluid replacement with ophthalmic viscosurgical devices, continuous curvilinear capsulorhexis with capsular forceps, hydrodissection and hydrodelineation with balance salt solution, nuclear disassembly with the divided and conquer technique, cortical removal, implantation of intraocular lens, removal of ophthalmic viscosurgical devices, and wound sealing with hydration. After the completion, intracameral cefuroxime 1mg/0.1cc was injected. Before pressure bandage with gauze and patching with metal shield, 0.5% levofloxacin eyedrops, and 0.3% tobramycin and 0.1% dexamethasone ophthalmic ointment were instilled.
Q6 Please explain what kind of study is presented. It seems like an observational type.
A6
This is indeed a prospective cohort study, longitudinally observing pre- and post-operative corneal optical density and thickness. We have revised the abstract (page 1, lines 17-20) and Materials and methods (page 2, line 76) to clarify the type of our study.
Q7 Please add the sample size calculation.
A7
The sample size was determined by a free online calculator (https://sample-size.net/sample-size-study-paired-t-test/) supported by the Clinical and Translational Sciences Institute of University of California, San Francisco. According to the prior result of corneal thickness change at the wound from the first qualified 10 patients, we estimated the sample size by adopting the significance level (α) as 0.05, the desired power (1-β) as 0.8, the effect size of 18, and the standard deviation of the corneal wound’s thickness change of 33.2. Accordingly, the estimated sample size was at least 29 subjects. We have added a statement about the sample size estimation in Materials and Methods (page 4, lines 135-142).
Q8 Please add axial length and anterior chamber depth in the demographic data table.
A8
We have added the axial length and anterior chamber depth in the demographic data in Table 1 and added a statement in Results (page 4, lines 154-155).
Q9 Please acknowledge the limitations of the current study such as non-randomisation, etc.
We thank the reviewer for this reminder. We have added a paragraph to acknowledge the limitations of this study in Discussion (page 9, lines 304-314). The limitation of this study is that the indices are measured only with Pentacam. We do not compare it with other Scheimpflug cameras or anterior segment optical coherence tomography currently available on the market. Whether the other Scheimpflug cameras present with the same pattern of densitometry and pachymetry needs further studies to elucidate. Another pitfall is difficulty in passing all of the automatic quality assessments of Pentacam AxL®, which led to the exclusion of many subjects initially recruited for analysis. Despite the instrument claiming to have a short measuring time within 2 seconds, this examination may still be difficult for certain patients. In elderly patients, there are many unpredictable conditions that may result in the challenge of performing this examination, such as achalasia, dry eye disease, hearing disability, hard to cooperate, which need experienced technicians to overcome.
Reviewer 2 Report
The idea of assessing the COD at different annuli is great. The evaluation of CT is not so original, but partially useful. But COD, at our Knowledge, is definitely intriguing. The study is well-written. Figures are accurate, even if a little bit complicated to quickly understand. Personally, I prefer Fig. 3 and 5 as graphical attempt to summarize the concept of the manuscript, that is comparing the whole CODs and CTs to obtain valuable indices of corneal injury after phacoemulsification. The pitfall of this study is that indices are measured only with Pentacam. To optimize the potential use of these indices on a large scale, they should be compared from each Scheimplug cameras currently available on the market.
Author Response
We appreciate the time and effort that the reviewer dedicated to providing feedback on this manuscript. The pitfalls of this study as the reviewer pointed out have been added to the discussion (page 9, lines 304-308). Thanks for the valuable suggestion and positive critique.